# Collagen-Based Osteogenic Nanocoating of Microrough Titanium Surfaces

**DOI:** 10.3390/ijms23147803

**Published:** 2022-07-15

**Authors:** Christina Behrens, Philipp Kauffmann, Nikolaus von Hahn, Uwe Schirmer, Klaus Liefeith, Henning Schliephake

**Affiliations:** 1Department of Oral and Maxillofacial Surgery, George-Augusta-University, 37075 Göttingen, Germany; cbehren@gwdg.de (C.B.); philipp.kauffmann@med.uni-goettingen.de (P.K.); nikolaus.von-hahn@med.uni-goettingen.de (N.v.H.); 2Institute for Bioprocessing and Analytical Measurement Techniques, 37308 Heiligenstadt, Germany; uwe.schirmer@iba-heiligenstadt.de (U.S.); klaus.liefeith@iba-heiligenstadt.de (K.L.)

**Keywords:** polyelectrolyte multilayer, heparin, collagen, bone morphogenic proteins, controlled release, recombinant proteins, biofunctionalization

## Abstract

The aim of the present study was to develop a collagen/heparin-based multilayer coating on titanium surfaces for retarded release of recombinant human bone morphogenic protein 2 (rhBMP2) to enhance the osteogenic activity of implant surfaces. Polyelectrolyte multilayer (PEM) coatings were constructed on sandblasted/acid-etched surfaces of titanium discs using heparin and collagen. PEM films of ten double layers were produced and overlayed with 200 µL of a rhBMP2 solution containing 15 µg rhBMP2. Subsequently, cross-linking of heparin molecules was performed using EDC/NHS chemistry to immobilize the incorporated rhBMP2. Release characteristics for 3 weeks, induction of Alkaline Phosphatase (ALP) in C2C12 cells and proliferation of human mesenchymal stem cells (hMSCs) were evaluated to analyze the osteogenic capacity of the surface. The coating incorporated 10.5 µg rhBMP2 on average per disc and did not change the surface morphology. The release profile showed a delivery of 14.5% of the incorporated growth factor during the first 24 h with a decline towards the end of the observation period with a total release of 31.3%. Cross-linking reduced the release with an almost complete suppression at 100% cross-linking. Alkaline Phosphatase was significantly increased on day 1 and day 21, indicating that the growth factor bound in the coating remains active and available after 3 weeks. Proliferation of hMSCs was significantly enhanced by the non-cross-linked PEM coating. Nanocoating using collagen/heparin-based PEMs can incorporate clinically relevant amounts of rhBMP2 on titanium surfaces with a retarded release and a sustained enhancement of osteogenic activity without changing the surface morphology.

## 1. Introduction

Enhancement of the integration of endosseous implants into surrounding bone through coatings using bioactive organic molecules has been approached in various ways) [1,2]. Among the biomacromolecules used, collagen has been widely explored as an ubiquitous component of the extracellular matrix with a positive effect on cell adhesion, proliferation and migration. The previous work of our group had used nano-anchorage, direct adsorption and biomimetic assembly for fixation of collagen chains to the metallic surface of both smooth and microrough titanium implants with a positive effect on peri-implant bone formation when compared to uncoated surfaces [3,4,5,6,7,8,9,10]. More recently, implants with dual-etched, collagen-coated surfaces have confirmed the enhancing effect of this type of coating on osseointegration [11,12,13]. Binding of collagen to titanium and titanium alloy surfaces has also shown to improve soft tissue biocompatibility in vitro and in vivo [14,15,16,17].

Biologically active molecules such as RGD peptides and rhBMP2 had been introduced into the coating layer using covalent binding and adsorption to the collagen fibres in a biomimetic approach [4,6,7,8,9]. This, however, did not lead to a further enhancement of bone integration. More recent approaches to collagen coatings in conjunction with bioactive molecules have used a stepwise layer-by-layer strategy alternating collagen with heparin layers to compose a multilayer coating of expanded polytetrafluoroethylene (ePTFE) stents in combination with RGD peptides [18] and of collagen scaffolds for tissue engineering applications in conjunction with basic fibroblast growth factor (bFGF) [19]. The strategy of sequentially combining negatively charged heparin layers with weak cationic collagen molecules in a polyelectrolyte multilayer (PEM) coating may allow for a more controlled coating composition than the previously employed biomimetic approach that had used a spontaneous assembly of collagen, glycosaminoglycans (GAG) and growth factors [7,8]. Moreover, the possibility to manipulate the structure of the multilayer system by sequential or intermittent cross-linking may allow for more effective experimental variation and modulation of its biological activity. To the best knowledge of the authors, multilayer coatings using heparin and collagen on titanium surfaces for the enhancement of osteogenic properties have not yet been reported.

It was thus the aim of the present study to develop a collagen/heparin-based multilayer coating on titanium surfaces to enhance the biological performance of implant surfaces. Loading of rhBMP2 in conjunction with targeted manipulation of the multilayer structure was assessed to modify the in vitro characteristics of growth factor release and the resulting differences in biological performance of osteogenic cell populations.

## 2. Materials and Methods

### 2.1. Specimen Fabrication

Commercially pure titanium (Ti) discs of 14.7 mm diameter were prepared by sandblasting and subsequent acid-etching (KLS Martin, Tuttlingen, Germany). The samples were etched in 5.1 M hydrochloric acid and 4.6 M sulphuric acid solution for 300 s by 108 °C [20]. 

### 2.2. Collagen Multilayer Coating of Tidiscs

The surfaces of the Tidiscs were coated in two different approaches: (i) Multilayer Collagen-Heparin (Col-Hep) coating loaded with rhBMP2; and (ii) Multilayer Col-Hep coating with different degrees of cross-linking and variation in the sequence of growth factor loading.

#### 2.2.1. Multilayer Col-Hep Coating

Poly-L-lysine (PLL) and Heparin (Hep) were purchased from Sigma Aldrich (Taufkirchen, Germany) and used without further purification, unless stated otherwise. The collagen type I was purchased from ibidi (Graefelfing, Germany, rat tail collagen type I, 5 mg/mL). The Col-Hep PEM films were assembled using PLL (30–70 kDa and Heparin (50 mg/mL, from porcine intestinal mucosa) as the initial layer. After assembling the first (PLL-Hep) double layer, nine (Col-Hep) double layers were added to the surface. The polyelectrolytes were dissolved in 5 mM acetate at a concentration of 1 mg/mL. Film construction was performed semi-automatically employing a dipping robot (DR3, Riegler&Kirstein, Potsdam, Germany). Briefly, the cleaned substrates were first soaked into the polycation solution (PLL) and left there for 5 min until an adsorption equilibrium was established. Subsequently, the samples were soaked into three deionized water wash solutions to rinse the surface to remove unbound polyelectrolytes. The polyanion Hep was adsorbed likewise by incubation for 5 min followed by three rinsing steps. After the first double layer, the polykation solution was switched to a 1 mg/mL collagen solution and the dipping protocol was changed. For the Hep-Collagen film construction, the dipping time in each polyelectrolyte was increased to 30 min followed by three rinsing steps. The film construction was performed by repeating these cycles until reaching the desired number of double layers and film architecture, respectively. All samples were rinsed in deionized water and air dried.

Surface morphology was characterized using (i) field emission scanning electron microscopy (FESEM; ULTRA plus, Carl Zeiss, Jena, Germany) with carbon sputtered specimens (EM ACE200, Leica, Wetzlar, Germany) and (ii) profilometry (Smartproof 5, Carl Zeiss, Jena, Germany) in conjunction with automated software analysis (ConfoMap ST, vers. 7.4.8076, Carl Zeiss). The uncoated Ti surfaces exhibited the typical pits-and-grooves pattern, well-known from sandblasted, acid-etched surfaces. The nanocoating with PEM multilayer films and growth factor loading did not alter the morphologic characteristics of the etched and sandblasted surface of the Ti specimens (Figure 1A–C). Surface roughness (R_a_/S_a_) of the Ti specimens with/without PEM films with/without loading with rhBMP2 varied between 3.4101 µm/3.6249 µm and 4.4159 µm/4.8345 µm and 4.1287 µm/4.5938 µm without statistical significance (*p* < 0.223 and 0.083, respectively) (Figure 1D–F).

#### 2.2.2. Growth Factor Loading

Previous experiments had shown that solutions with 75 µg rhBMP2/mL had been appropriate to adequately load PLL-Heparin multilayer systems on Ti surfaces [21]. For the multilayer systems under evaluation in the present study, the (PLL-Hep)(Col-Hep)_9_ multilayer system, and 200 µL of loading solutions with 75 µg rhBMP2/ml (CHO cells, PeproTech, Hamburg, Germany) were used, corresponding to 15 µg per specimen. PEM-coated titanium samples were each deposited into a single well of a 24-well plate and incubated over night at 4 °C with 200 µL (15 µg) of the BMP2. After loading, the supernatant was transferred to a reaction tube for further use and the discs were washed twice with deionized water and shortly air-dried at RT. For the covalent immobilization of rhBMP2 to the PEM, cross-linking was performed as described below.

The amounts of rhBMP2 bound to the multilayer systems were determined indirectly from the supernatant of the coating procedure. Growth factor concentrations in the supernatant were assessed using the Bicinchoninic Acid (BCA) Protein Assay Kit (ThermoScientific, Darmstadt, Germany). Bovine serum albumin (BSA) was used as standard. In total, 25 µL of each standard (working range 25–2000 µg/mL or 5–250 µg/mL) and unknown sample were replicated into a microplate well (96-well plates). Then, 200 µL of working solution was added to each well, the plates were mixed thoroughly on a plate shaker for 30 s (37 °C, 400 rpm, THERMOstar, BMG LABTECH, Ortenberg, Germany) and incubated at 37 °C for 30 min. Absorbance was measured with an ELISA plate reader (SpectraMax M2, Molecular Devices, San Jose, CA, USA) at 562 nm, room temperature (RT).

#### 2.2.3. Cross-Linking

Cross-linking was done using EDC/NHS (1-ethyl-3-(3-dimethylaminopropyl) carbodiimide/N-hydroxysuccinimide; abcr Gmbh, Karlsruhe, Germany) at three concentration levels. The number of possible binding sites for cross-linking were calculated and coverage of 1%, 10% and 100% of theoretically available binding sites were addressed using 0.077, 0.77 mg/mL and 7.7 mg/mL EDC to achieve cross-linking of the Col-Heparin multilayer system after the growth factor molecules had been loaded. As described previously [21], the EDC as well as the NHS was diluted in ice cold 0.15 M NaCl solution pH 5. Whereas the EDC concentration could vary as described above, the final NHS concentration was constantly 11 mg/mL for each sample. The cross-linking reaction was conducted over night at 4 °C. After the cross-linking, all samples were washed at RT three times with 0.15 M NaCl, pH 8. Between each washing step, the samples were incubated for 1 h. Finally, the samples were rinsed with water and allowed to air dry at RT. The cross-linking of the PEMs was performed after cytokine loading for the covalent immobilisation of cytokines to the PEM films.

### 2.3. Release Experiments

All coated titanium discs were placed into 24-well plates and incubated in 1 mL DMEM supplemented with 2% FCS and 1% penicillin/streptomycin at 37 °C in a 5% CO_2_ atmosphere and shaken at 70 rpm (Celltron, InforsHT, Einsbach, Germany). The medium was collected and replaced after 24, 48, 72 h and every 3 days thereafter until day 21. The supernatants were stabilized with protease inhibitor (ROCHE Diagnostics, Mannheim, Germany). The release profiles of rhBMP2 were assessed using a Human/Murine/Rat BMP2 TMB ELISA Development Kit (PeproTech, Hamburg, Germany), according to the instructions of the supplier. Briefly, 100 µL of each unknown sample (supernatant from release experiments with different amounts of diluent) and standard (rhBMP2, PeproTech) were added to a microtiter plate coated with anti-BMP2 (overnight at RT, 0.25 µg/mL) and incubated for 2 h at RT and 400 rpm on a plate shaker (THERMOstar). After several washing steps, a BMP2 detection antibody (0.25 or 0.5 µg/mL) was added followed by incubation for 2 h at RT and 400 rpm. After incubation with Streptavidin-HRP (30 min, RT, 400 rpm, 0.05 or 0.1 µg/mL), the TMB substrate solution was added to each well and incubated at RT and 400 rpm for color development for 20 min. Reaction was stopped with 1 M HCL stop solution and color development was measured with an ELISA plate reader (SpectraMax M2) at 450 nm with wavelength correction set at 620 nm. All measurements were done in duplicate on three specimens for each surface condition.

### 2.4. Osteogenic Activity of Surface Conditions

Osteogenic activity associated with the different surface conditions were evaluated in C2C12 cells (DSZM, Braunschweig, Germany) and human mesenchymal stem cells (hMSCs) (Lonza, Verviers, Belgium). In C2C12 cells, induction of Alkaline Phosphatase (ALP) was assessed both in cells grown separately on tissue plastic using the supernatant released from the surfaces and in C2C12 cells grown on the surface of the Tidiscs. In hMSCs, the proliferation rate was assessed. Cells were cultivated at 37 °C in a 5% CO_2_ atmosphere. C2C12 cells and hMSCs were grown in Dulbecco’s Modified Eagle´s Medium GlutaMAX (DMEM; ThermoFisher, GIBCO, Darmstadt, Germany) supplemented with 10% fetal calf serum (FCS; Biochrom, Berlin, Germany) and 1% penicillin/streptomycin (PAN Biotech, Aidenbach, Germany). The medium was exchanged every 3–4 days after washing the cells with PBS. Until reaching 70–80% confluency, the cells were detached from the culture flask surface using a trypsin/EDTA solution (0.25% trypsin/0.02% EDTA in 1x PBS; PAN Biotech) and subcultivated.

For the evaluation of the supernatant, cells were seeded at a density of 40,000 cells/well in 24-well plates and cultivated in DMEM with 2% FCS and 1% penicillin/streptomycin. After attachment of the cells, supernatants from the release experiments were added at concentrations of 50 ng/mL rhBMP2, which was identified as the threshold concentration of rhBMP2 (Behrens et al. 2022). To analyze the BMP2 activity of the multilayer coatings on the titanium discs, 40,000 cells were seeded onto each disc in 24-well-plates and cultivated in DMEM with 2% FCS and 1% penicillin/streptomycin. The measurement of ALP was carried out after 72 h, whereby the titanium discs were transferred into new 24-well plates. The cells were washed with PBS and lysed with 300 µL lysis buffer/well for 1 h at RT on a plate shaker (THERMOstar) at 400 rpm. The lysis buffer was composed of 0.1 M glycin, 1 mM MgCl_2,_ 1 mM ZnCl_2_ and 1% NonidetP-40 (Octylphenoxypolyethoxyethanol). The lysates were centrifuged at 1500 rpm (Eppendorf benchtop centrifuge 5415) for 5 min at RT and 75 µL of each supernatant was transferred in duplicate into a 96-well plate. After addition of 75 µL substrate buffer containing 2 mg/mL P-Nitrophenyl phosphate (pNPP; SigmaAldrich Chemie, Taufkirchen, Germany), the samples were incubated at 400 rpm on a plate shaker (THERMOstar) at 37 °C. Absorbance was measured at 405 nm and 37 °C at 60 and 90 min. (SpectraMax M2).

The influence of rhBMP2 on the proliferation and viability of cells was determined using the colorimetric CellTiter 96^®^ AQ_ueous_ One Solution Cell Proliferation Assay (Promega, Madison, WI, USA). The assay solution contains the tetrazolium compound 3-(4,5-dimethylthiazol-2-yl)-5-(3-carboxymethoxyphenyl)-2-(4-sulfophenyl)-2H-tetrazoli-um (MTS) combined with the electron coupling reagent phenazine ethosulfate (PES). The amount of formazan product formed, as indicated by the absorbance, was proportional to the number of viable, metabolically active cells. Hence, changes in the metabolic activity of the cell population was used as a surrogate parameter for changes in cell numbers, i.e., cell proliferation.

For the assessment of proliferation, 30,000 hMSCs were seeded onto coated titanium discs in 24-well plates and cultivated in DMEM with 10% FCS and 1% penicillin/streptomycin for 72 h. The discs were transferred into new 24-well plates and, after washing the cells with PBS, 500 µL fresh culture medium and 100 µL assay solution per well were added to measure the amount of soluble formazan produced by cellular reduction of MTS. After incubation for 2 h at 37 °C in a 5% CO_2_ atmosphere, 100 µL aliquots from each well were transferred into a 96-well plate in duplicate and the absorbance was measured at 490 nm with an ELISA plate reader (SpectraMax M2). Three discs from each group were evaluated at each interval with all measurements performed in duplicate.

### 2.5. Statistics

Data are presented as means ± standard deviation (SD). Univariate ANOVA (SPSS Statistics 24.0), were used to compare the release kinetics between the three different surfaces. Univariate ANOVA with Bonferroni correction was employed to compare growth factor release, induction of ALP through released growth factor and direct contact with the PEM films as well as proliferation of hMSCs. Univariate ANOVA with Bonferroni correction was also used to compare the R_a_ values of the uncoated/coated Ti specimens.

## 3. Results

### 3.1. Release of Growth Factors

#### 3.1.1. Col-Hep Multilayer Systems

Uncoated control titanium discs were adsorbed 2.1 µg rhBMP2 (SD 0.3) and released 19.6 % (SD 5.3) within the observation period of 21 days. The unmodified Collagen/Heparin multilayer films had incorporated 10.5 µg of rhBMP2 on average (SD 2.4). The release profile exhibited a rapid release of 1411.7 ng rhBMP2 (SD 495.2) within one day, after which a much slower release occurred. After 21 days, 3066.4 ng (SD 728.3) had been delivered, which corresponded to 31.3 % of the loaded growth factor (Figure 2A,B).

#### 3.1.2. Cross-Linked Col-Hep Multilayer Systems

The release profile of the films that had been cross-linked after loading with rhBMP2 was similar to that of the non-cross-linked films with the highest release on day 1 and a subsequent flattening of the release curve (Figure 2A). The total release from these multilayer films was 21.8% (SD 20.7) for the films with 1% cross-linking. Films with 10% cross-linking had reduced the release to 3.7% (SD 1.9), whereas 100% cross-linking had entirely suppressed the release to 0.0% of the loaded BMP2 (Figure 2B). Differences between the cross-linked and the unmodified films were significant (*p* < 0.000).

### 3.2. Osteogenic Activity

#### Induction of ALP through Unmodified and Cross-Linked Multilayer Systems

Osteogenic activity of the released rhBMP2 from the multilayer films as assessed by induction of ALP in C2C12 cells grown separately on tissue plastic showed a significantly increased ALP-activity in cells exposed to the released rhBMP2 from the non-cross-linked films as compared to the negative controls. RhBMP2 released from the EDC cross-linked multilayer films induced a decreased activity with the level of induction decreasing with increasing degree of cross-linking that was still significantly enhanced (1%: *p* < 0.001, 10%: *p* < 0.018). Col-Hep multilayer films with 100% cross-linking could not be evaluated due to the lack of released rhBMP2 (Figure 3A).

Unmodified multilayer films and those that had been cross-linked induced significant amounts of ALP in C2C12 cells grown on the respective surfaces at the start of the observation period at day 1 (Figure 3B). After 21 days of release, all the films loaded with rhBMP2 still induced significantly enhanced levels of ALP-activity (*p* < 0.000).

### 3.3. Cell Proliferation

Human MSCs exhibited a significantly increased rate of proliferation on unmodified multilayer systems loaded with rhBMP2. The cross-linked films showed a significant difference in hMSC proliferation only on the 1% cross-linked surfaces (*p* < 0.001), whereas films with higher degrees of cross-linking did not induce a higher proliferation rate than the controls. Interestingly, the collagen containing multilayer films without growth factor loading had a significantly enhancing effect on the proliferation of hMSCs compared to uncoated Ti surfaces (*p* < 0.000) (Figure 4).

## 4. Discussion

The present study has used collagen I in combination with heparin in a layer by layer approach to establish a polyelectrolyte multilayer film on titanium surfaces that can harbour growth factors to enhance the osteogenic potential of the metal surface. Heparin is known to provide binding sites for many polypeptide growth factors [22] and many growth factors involved in bone regeneration themselves have specific binding sites for sulfate groups of the heparin molecule [23,24]. Collagen in turn provides a number of binding sites for cells and thereby offers attractive conditions for cell attachment and migration. The assembly of the two components on the metallic surface during the layer-by-layer approach was expected to allow for a more controlled and effective way of accommodating growth factors in the resulting films than with previously used self-assembly of collagen and glucosaminoglycans together with BMPs [6,8,25]. The amount of rhBMP2 that has been incorporated into the multilayer films on the surface corresponds to or even exceeds the amounts of BMP previously reported to be incorporated into multilayer on Ti surfaces using heparin or hyaluronic acid in conjunction with poly-L-lysin or chitosan as cationic partners depending on the concentration of the loading solution [21,26,27]. The amount of rhBMP2 released from collagen–heparin multilayer films has shown to be by three orders of magnitude higher than previously reported for self-assembled Collagen-Proteoglycan films loaded with BMP2 [25] and can be expected to elicit a stronger biological reaction compared to the previously tested self-assembled coatings on Ti implants [7].

Cross-linking appears to play a crucial role in the biological activity of surface coatings using PEMs. Covalent cross-linking of collagen-hyaluronic acid multilayer films on Ti surfaces has provided increased chemical stability and improved the biological performance of stem cells grown on these films using the EDC-NHS system [28,29]. This system has been employed using rather large amounts in PLL-Hyaluronic Acid PEMs before loading of rhBMP2 in order to modify the release profile. However, a previous study on PLL-Heparin films has shown that larger amounts of EDC substantially reduced the biological activity of the incorporated BMP and that applying small amounts of EDC after growth factor loading more effectively immobilized the loaded growth factor [21]. Thus, the present study used the same approach, but even the small amounts of EDC used had reduced the biological activity of the released rhBMP2 showing lower levels of induction of ALP in C2C12 cells with increasing degrees of cross-linking of the films. This may have occurred due to unbound EDC that could have interacted with the carboxylic or primary amine groups of the growth factor and thereby reduced its activity at the receptor binding site. Induction of ALP-activity in cells grown on the Col-Hep-BMP2 films surface was high on day 1 in non-cross-linked Col-Hep films and in those with cross-linking of 1% and 10%. After 21 days of release, the rhBMP2 remaining in these films maintained the ability to induce ALP in cells growing on the surface despite a negligible residual release. This fact has also been observed with C2C12 cells grown on BMP-loaded PLL-Hyaluronic Acid PEMs and it has been speculated that cells on the surface of the PEM films make growth factor molecules available by degradation of the film immediately underneath the cells through enzymes, such as metalloproteases and hyaluronidases [30]. On the contrary, in the present study, the films with 100% cross-linking did neither release BMP in measurable amounts nor induce ALP in cells growing on the surface, indicating that all incorporated growth factor molecules were completely immobilized after incorporation into the films and were also inaccessible for cells grown on the surface, rendering this modification biologically inactive.

The percentage of release of rhBMP2 from unmodified Col-Hep PEMs is significantly higher than the release from pure Ti surfaces, which may question the approach using these PEM films as to retard growth factor delivery from implant surfaces. The reservoir function, however, that is provided by the multilayer films allows for the accommodation of a much higher absolute amount of rhBMP2 in the PEM films, resulting in a much higher and more durable biological activity than the bare Ti surface loaded with BMP2. Moreover, the biological activity of the rhBMP2 adsorbed to Ti surfaces had shown significantly lower levels of biological activity in ALP induction. One may speculate that this is not only due to the much higher amount of growth factor incorporated in the PEM films but also due to a different way of presentation of the BMP2 molecules to the cells growing on the PEM films vs. bare Ti surfaces.

For cell attachment and proliferation, the biological value of the use of a Col-Hep multilayer film as such can be seen when compared to uncoated Ti. Here, the unloaded collagen-heparin multilayer films had increased the proliferation of stem cells to a level that was comparable to the biological activity of uncoated Ti surfaces with adsorbed rhBMP2. Indirectly, this is well-known from previous preclinical in vivo studies reporting a significantly better performance of collagen-coated surfaces with respect to peri-implant bone formation [7,8,31]. In these reports, addition of rhBMP2 to the surface coating failed to show any additional effect on peri-implant bone formation, probably due to the low amount of BMP2 incorporated or the low release of the growth factor from these coatings. The present approach of a controlled layer-by-layer composition has shown that the addition of rhBMP2 to collagen/heparin-based multilayer films in a nanocoating approach has resulted in a significant enhancement of the in vitro performance of the BMP2-loaded surface films by increasing the proliferation rate of hMSCs. This difference is levelled off by covalently immobilizing the incorporated growth factor by cross-linking using EDC, which is also reflected by the decreased release of rhBMP2 from PEM films cross-linked to 1% and 10%. The reduction in proliferation of hMSCs in cross-linked Col-Hep films without rhBMP2 loading may be accounted for by the fact that cell-binding motifs such as RGDs in collagen molecules have been altered or were less available after EDC/NHS cross-linking.

The negative effect of EDC on the efficacy of the released rhBMP2 in conjunction with the reduced enhancing effect on proliferation of hMSCs suggests that cross-linking of Col-Hep-PEM films does not contribute to an improved biological performance of rhBMP2. Moreover, a recent study reported increased inflammatory in vivo reactions after the use of EDC/NHS cross-linking in conjunction with collagen [32], which may be detrimental for peri-implant bone formation.

In conclusion, the present study has shown that nanocoating of microrough Ti surfaces using collagen based multilayer films can accommodate rhBMP2 in the µg range on the surface without altering the surface morphology and can maintain the biological activity of the released growth factor for several weeks. Thereby, the biological in vitro performance of the implant surface itself is significantly improved with respect to osteogenic activity and induction of osteogenic markers of cells grown on these surfaces and by enhancing the proliferation of human mesenchymal stem cells. Cross-linking of PEMs in order to retain higher amounts of growth factor in the films appears to reduce the biological performance.

## Figures and Tables

**Figure 1 ijms-23-07803-f001:**
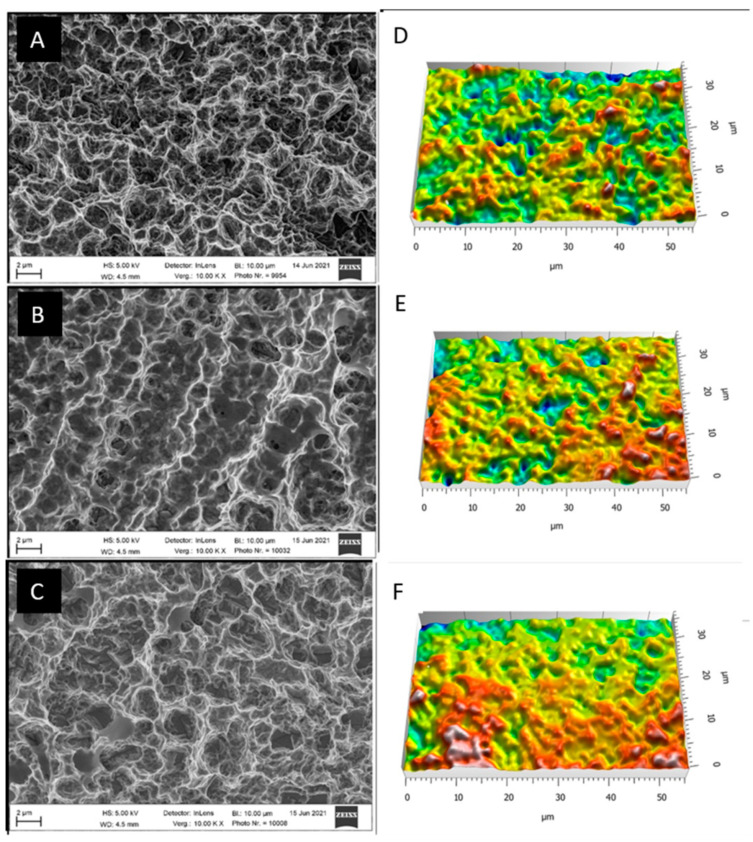
Surface Characterization: (**A**): SEM image of uncoated sandblasted, acid-etched Ti surface. (**B**): SEM image of Ti surface coated with (PLL-Hep)(Col-Hep)_9_ PEM. (**C**): SEM image of Ti surface coated with (PLL-Hep)(Col-Hep)_9_ PEM, loaded with 15 µg rhBMP2 (R_a_: 4.1287 µm, SD 0.7171, S_a_: 4.5894 µm, SD 0.5226). (**D**): Profilometric image of uncoated sandblasted, acid-etched Ti surface (Mena values: R_a_: 3.4101 µm, SD 0.4346, S_a_: 3.6249 µm, SD 0.6603). (**E**): Profilometric image of Ti surface coated with (PLL-Hep)(Col-Hep)_9_ PEM, (Mean values: R_a_: 4.4159 µm, SD 0.7359, S_a_: 4.8345 µm, SD 0.4876). (**F**): Profilometric image of Ti surface coated with (PLL-Hep)(Col-Hep)_9_ PEM, loaded with 15 µg rhBMP2 (Mean values: R_a_: 4.1287 µm, SD 0.7171, S_a_: 4.5894 µm, SD 0.5226). Comparison of results of the profilometric analysis of the three surface conditions revealed no significant differences (R_a_: *p* < 0.223, S_a_: 0.083).

**Figure 2 ijms-23-07803-f002:**
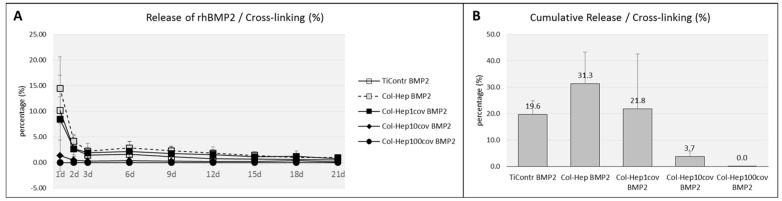
(**A**): Release profile of rhBMP2 from films with different degrees and modes of cross-linking. (**B**): Cumulative percentage of total release of rhBMP2 from PEM films. Data are presented as means ± standard deviation (SD) of duplicate measurements of *n* = 3 specimens each.

**Figure 3 ijms-23-07803-f003:**
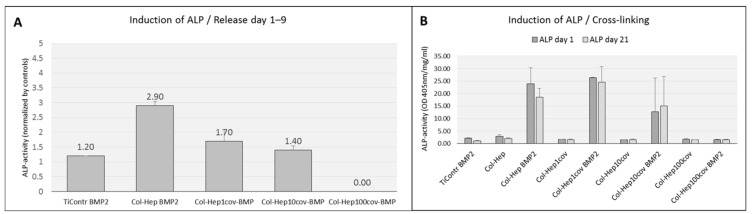
(**A**): Induction of ALP in C2C12 cells grown on tissue plastic by the released rhBMP2 from (PLL-Hep)(Col-Hep)_9_ films compared to control cells. (**B**): Induction of ALP in C2C12 cells grown on (PLL-Hep)(Col-Hep)_9_ coated Ti surfaces with different degrees of PEM cross-linking. Data are presented as means ± standard deviation (SD) of duplicate measurements with *n* = 2 (**A**) and *n* = 3 (**B**) specimens each.

**Figure 4 ijms-23-07803-f004:**
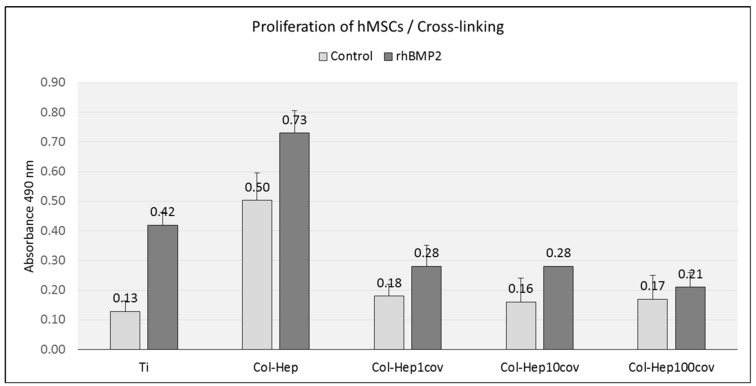
MTS assay for hMSCs after 3 days of culture on (PLL-Hep)(Col-Hep)_9_ covered Ti surfaces with different degrees of PEM cross-linking. The number of viable cells on the different surfaces was determined by measuring absorbance at 490 nm, which is directly proportional to the number of metabolically active cells. Data are presented as means ± standard deviation (SD) of duplicate measurements of *n* = 3 specimens each.

## Data Availability

Not applicable.

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
