# Peer review of "Collagen-Based Osteogenic Nanocoating of Microrough Titanium Surfaces"

_ijms, 2022, doi:10.3390/ijms23147803_

Round 1
Reviewer 1 Report
The paper entitled “Collagen based bioactive nanocoating of microrough Ti-surfaces for growth factor release enhances osteogenic properties’’ presents a topic of real interest regarding biomaterials with biomedical applications. The manuscript presents nanocoating of microrough Ti-surfaces using collagen/heparin based multilayer films for retarded release of bone morphogenic protein 2 (rhBMP2). The enhancement of osteogenic activity without changing the surface morphology is obtained.
The paper is very well written, easy to understand, offering potentially important information. The following concerns should be addressed before the manuscript can be considered for publication.
· The title of the paper should be more concise.
· “in vitro” and “in vivo” must be written in Italic. Please, make the correction throughout the entire manuscript.
· Taking into account that the cross-linking of PEMs plays an important role in retaining of growth factor in the films, and on the biological performance, the authors should demonstrate, by FTIR analysis, the surface chemistry of the prepared PEM films with different degrees and modes of cross-linking.
· The authors refer that cross-linking by EDC/NHS was done at three concentration levels. Did the authors test the PEM films cross-linked with more than 1% and less than 10%?
· In the manuscript, the authors refer to previous experiments, but those citations do not appear in the References. (Please see, PP. 4, Line 123 ‘’Previous experiments had shown that solutions with 75 µg rhBMP2/ml had been appropriate to adequately load PLL-Heparin multilayer systems on Ti surfaces (Behrens et 123 al. 2022).’’ PP. 4, Line 148 ‘’As described previously (Behrens et al., 2021)’’.
Overall, this manuscript is valuable and deserves a publication after major revision.
Reviewer 2 Report
The authors have written a well-referenced manuscript describing the effect of crosslinking on growth factor release from Ti surfaces. If the effects are as significant as they indicate, then the paper is a timely and interesting step forward to understanding the coating of implants. I do however have concerns with the small number of repeats used.
The methods state (pg 5 line 6) that the release study was repeated twice. This may have contributed to the very high SD in Fig 2B, which provides little confidence in the comparisons of samples. While the results appear logical such a large variance undermines the confidence of this work. “The total release from these multilayer 240 films was 21.8% (SD 20.7) for the films with 1% crosslinking. Films with 10% cross-linking 241 had reduced the release to 3.7% (SD 1.9)”
Line 271-272: Can you comment on why this increase in cell metabolism is occurring on CollHep but not on any other material? How confident are you this result is correct with only 3 repeats?
Why are the positive controls not shown in figure 3?
Include details of the test used in Fig 4 legend.
Include details on the numbers of repeats/experimental units in figure legends.
p=0.000 is an unusual convention for showing P values. Perhaps consider reporting the exact P-value, or state P<0.001/0.0001? “On SPSS, Double-click the output table. Select the cells containing p-values. Right-click Cell Properties and adjust the number of decimal places to report the exact p-value”
It is a little pedantic, but are you not strictly speaking measuring cell metabolic activity and not cell proliferation in figure 4?
Round 2
Reviewer 1 Report
The manuscript deserves publication in the present form.
Reviewer 2 Report
I thank the authors for addressing my comments with their study, I would now recommend the publication of the manuscript.